# Assessment of Adherence to the Healthy Food Pyramid in Pregnant and Lactating Women

**DOI:** 10.3390/nu13072372

**Published:** 2021-07-11

**Authors:** Andrea Gila-Díaz, Ariadna Witte Castro, Gloria Herranz Carrillo, Pratibha Singh, William Yakah, Silvia M. Arribas, David Ramiro-Cortijo

**Affiliations:** 1Department of Physiology, Faculty of Medicine, Universidad Autónoma de Madrid, C/Arzobispo Morcillo 2, 28029 Madrid, Spain; andrea.gila@uam.es (A.G.-D.); ariadna.witte@estudiante.uam.es (A.W.C.); silvia.arribas@uam.es (S.M.A.); 2Division of Neonatology, Hospital Clínico San Carlos, Instituto de Investigación Sanitaria del Hospital Clínico San Carlos (IdISSC), C/ Profesor Martin Lagos s/n, 28040 Madrid, Spain; gherranz@gmail.com; 3Division of Gastroenterology, Beth Israel Deaconess Medical Center, Harvard Medical School, 330 Brookline Avenue, Boston, MA 02215, USA; psingh6@bidmc.harvard.edu; 4Department of Neonatology, Beth Israel Deaconess Medical Center, Harvard Medical School, 330 Brookline Avenue, Boston, MA 02215, USA; wyakah@bidmc.harvard.edu

**Keywords:** breastfeeding, diet, healthy food pyramid, nutrition, pregnancy

## Abstract

There are numerous dietary recommendations during pregnancy. However, there are limited recommendations during the lactation period, a nutritionally vulnerable period for women. The Mediterranean Diet and adherence to the Healthy Food Pyramid (HFP) is considered as the standard for healthy eating. In this study, we investigated the differences in adherence to the HFP in pregnant, lactating, and non-pregnant/non-lactating (NPNL) women concerning sociodemographic factors. A sociodemographic and nutritional and lifestyle questionnaire (AP-Q) were used to assess adherence to the HFP, including lifestyle. The AP-Q score ranges from 0 to 10 meaning the higher the score, the greater the adherence to the HFP. Lactating women had the lowest AP-Q score (6.13 [5.31; 6.82]) compared to the pregnant (6.39 [5.56; 7.05]) and NPNL women (6.27 [5.43; 6.88]), while pregnant women had the highest scores. Maternal age was positively correlated with AP-Q score in pregnant (rho = 0.22; *p*-Value < 0.001) and lactating women (rho = 0.18; *p*-Value < 0.001), but not in NPNL women. Educational level and monthly income had a positive influence on the degree of adherence to the HFP. In conclusion, breastfeeding mothers of young age and low socioeconomic and educational level would be the target population to carry out nutritional interventions that improve their adherence to the HFP. The knowledge gained from this study can help to design recommendation guidelines and nutritional educational interventions for a given population.

## 1. Introduction

Healthy lifestyle habits are crucial in preventing cardiometabolic diseases, as these diseases are widespread among the population. The World Health Organization (WHO) has identified unhealthy diets, low physical activity, alcohol, and tobacco consumption as the main modifiable factors contributing to the increase in chronic non-communicable diseases [1]. The role of lifestyle and nutrition in the health of the offspring has been well demonstrated [2] and are especially important during pregnancy and lactation. In Spain, there are nutritional guidelines for pregnant women, based on the recommendations of the Spanish Society for Community Population (SENC; [3]). However, the breastfeeding period lacks specific guidelines. Available guidelines only suggest that the diet during the lactation period should resemble pregnant women’s diet. However, the nutritional recommendations in the Clinical Practice Guideline for Breastfeeding [4] are mainly focused on the benefits for the infant. Therefore, breastfeeding women have limited information, nutritional monitoring, and follow-up. This may subsequently increase the risk of nutritional deficiencies in the mother, and in the breastmilk, which can lead to nutritional deficits in the newborn. Recommendations during this period could be useful to ensure optimal maternal and infant health.

The Mediterranean Diet is largely considered as a gold standard pattern for healthy eating and has been widely associated with lower incidence of cardiometabolic diseases [5,6]. The Healthy Food Pyramid (HFP) is a simplified graphic representation of the Mediterranean Diet developed by the SENC [7] and has been used as a reference for different Mediterranean areas and cultures [8]. In addition to adherence to the Mediterranean Diet, the Mediterranean lifestyle also considers adequate hydration, practice of physical activity, and social nutrition involving family and friends [9]. Adherence to the Mediterranean Diet has well-recognized benefits in reducing obstetric complications such as gestational diabetes and pregnancy hypertension [10,11]. Additionally, it also benefits the newborn due to the influence of maternal diet during fetal development [12]. Knowing whether women have a good adherence to the HFP during the gestational and breastfeeding periods, and which are the nutritional deficient areas, would allow establishing adequate and personalized guidelines, accompanied by effective nutritional interventions in these relevant stages for the health of women and future generations.

Different tools have been developed to study the dietary patterns of the population, including 24-h food recall, frequency food questionnaire [13], and dietary history [14]. In nutritional epidemiology, it is necessary to evaluate dietary patterns along with lifestyle. For this purpose, some instruments have been developed, such as the Trichopoulou’s questionnaire [15], which have been widely used to study adherence to the Mediterranean Diet [9,16,17]. However, it is not free from criticism, particularly the use of arbitrary cut-off points [18]. Another used instrument is the KIDMED questionnaire [19]. However, this questionnaire does not explore individual dietary patterns in detail and focuses on the infant and juvenile populations [20,21]. We have developed an instrument to measure adherence to HFP, which takes into consideration dietary patterns, lifestyle, and healthy habits. This questionnaire, named Adherence to the Healthy Food Pyramid questionnaire (AP-Q), has been validated for the general population [22] and the breastfeeding women [23]. Using this instrument, we studied the adherence to HFP in the female population of childbearing age in Spain, including women in the gestation and lactation periods, as well as non-pregnant and non-lactating women. In this study, we compared these groups to identify whether their dietary pattern and lifestyle are following the recommendations of the HFP. We also assessed possible areas of deficiencies and analyzed the influence of sociodemographic factors’ differences in lifestyle according to several sociodemographic factors.

## 2. Materials and Methods

### 2.1. Study Design and Participants Recruitment

This is a cross-sectional study aimed to analyze adherence to HFP and the relationship with socio-demographic determinants in women. The survey was performed through online questionnaires. To evaluate HFP adherence, we used the self-reported questionnaire AP-Q. The questionnaire was preceded by the following information: age (years), weeks of gestation in pregnant women, weeks of postpartum in lactating women, comorbidities (obesity, hypertension, diabetes mellitus, cancer, mood disorders, and inflammation-related diseases), and sociodemographic characteristics: origin (Spanish/non-Spanish), educational level (middle school or lower, high school and university degree), income (shown in euros per month: <1000€; 1000–2500€, 2500–4000€, ≥4000€), and current employment status (studying, working, unemployed or on maternity leave). The average response time was 30 min.

All data were anonymously collected, and no IP addresses were recorded. Participation was voluntary, and women had the opportunity to end their questionnaire any time without saving any of their previous responses. Participants were recruited mainly through non-profit breastfeeding associations, social networks, and maternity-specific discussion forums. Terminology that might prompt response to social weaknesses was removed to prevent any recruitment bias. The questionnaire was administered in Spanish using the online tool SurveyMonkey (https://es.surveymonkey.com/, September 2020 to January 2021) and completed by 3497 participants.

The general inclusion criteria for data selection included internet access, Spanish language comprehension, and age range between 18–45 years. We used this criterion to match the age of childbearing age and to avoid adolescent pregnancies, which may not be representative of the population. Pregnant women and breastfeeding women within two years of breastfeeding based on WHO recommendations [24] were also included. Exclusion criteria were no response or typographical errors. Mothers of twin infants meeting these criteria were asked to complete only one survey to avoid duplication. The final number of women matching inclusion criteria was 1867 and were categorized as non-pregnant/non-lactating (NPNL; *n* = 957), pregnant (*n* = 437), and lactating (*n* = 473) women (Figure 1).

This study was conducted in accordance with the principles of the Declaration of Helsinki, with the approval of the Ethical Committee of Universidad Autónoma de Madrid (CEI-UAM/102-1948).

### 2.2. Measurement of Women Dietary Patterns and Lifestyle

Adherence to the Healthy Food Pyramid questionnaire (AP-Q). A self-administered tool validated in the Spanish adult population [22] and breastfeeding women during the first month of lactation [23] was used. The AP-Q estimates the degree of adherence to the HFP, which is a standard for countries in the Mediterranean area [25,26]. It measures the frequency of consuming foods from different categories during the last month, as well as other aspects of the HFP related to lifestyle and healthy habits.

The AP-Q consists of 27 multiple-choice questions. The responses given to each item were grouped into 10 different categories, including physical activity, healthy habits and culinary techniques, hydration, grains, seed and legumes, fruits, vegetables, oil type, dairy products, animal proteins, and snacks. The healthy habits category includes four dimensions (lifestyle, emotional balance, sleep hygiene, and culinary techniques), and the hydration category contains dimensions of water intake, soft drinks, wine and beers, and distilled beverages. Each category and dimension were scored on a scale of 0 to 1. The dimensions with soft drinks, wine and beers and distilled beverages were scored on a scale from −1 and 1. The overall AP-Q score ranges from 0 to 10. The higher the score, the greater the adherence to the HFP. In the HFP, the food categories at the bottom of the pyramid have positive scores and represent daily practice, including physical activity, healthy culinary techniques, adequate hydration, consuming whole grains, seed and legumes, fruit and vegetables; while the categories at the top (snacks or foods that are not included in HFP) have negative scores [7].

### 2.3. Statistical Analysis

Quantitative variables were expressed as the median and interquartile range [Q1; Q3] and qualitative variables were reported as the percentage (%) along with the sample size. Differences between groups were analyzed by the Kruskal-Wallis test for quantitative variables and by Fisher’s exact test for qualitative variables. When Kruskal-Wallis was significant, Dunn test pairwise comparison between groups was applied. The correlation in quantitative variables was tested by Spearman’s rho coefficient. To test the association between the AP-Q and pregnant or breastfeeding women, regression models were performed considering NPNL women as the reference group. The models were adjusted by maternal age (continuous), origin (Spanish/non-Spanish), educational level (categorical), income (categorical), employment status (categorical), and comorbidities (yes/no). Estimated beta (β) coefficients ± standard error and associated *p*-Value were extracted from the models.

All data analyses were performed using R software (version 3.5.2, R Core Team 2018) within RStudio (version 1.1.453, Rstudio, Inc., Vienna; Austria) using the *rio, dplyr, devtools*, and *compareGroups* packages. Plots were generated using *ggplot2* and *ggpubr* packages. Results were considered statistically significant for *p*-value < 0.05.

## 3. Results

### 3.1. Population Characteristics

Table 1 shows the sociodemographic characteristics of the population. The median gestational age of the pregnant women was 23.0 [14.0; 32.0] weeks and the median of breastfeeding of the lactating women was 31.0 [15.0; 52.0] weeks postpartum. Maternal age was significantly higher in the breastfeeding group compared to pregnant or NPNL groups. More than 89.0% (1673/1867) of women were from Spain. In our cohort, 66.8% (1248/1867) had a university degree, followed by high school studies (25.4%) and few had only primary studies (10.0%). 53.8% (1004/1867) of women had a monthly income between 1000–2500 €; a high percentage of women in the NPNL and lactating groups were employed (55.0% and 38.7%, respectively), while pregnant and lactating women were on maternity leave (40.2%). In terms of health status, the prevalence of chronic disease was similar in pregnant and lactating women, being lower than in NPNL women.

### 3.2. AP-Q Score and Association between Women

Overall, the mean AP-Q score of the cohort was 6.09 ± 1.12. Pregnant women had significantly higher AP-Q score (6.39 [5.56; 7.05]) compared to the NPNL (6.27 [5.43; 6.88]) and lactating groups (6.13 [5.31; 6.82]). Lactating women had the lowest score (Figure 2A). The AP-Q score was significantly lower in non-Spanish (5.32 [4.57; 6.18]) than in Spanish women (6.34 [5.56; 6.95]; *p*-value < 0.001). The AP-Q score increased with educational level (Figure 2B), with income (Figure 2C), and in employed women (Figure 2D).

Maternal age and AP-Q score showed a positive and significant correlation for pregnant (rho = 0.22; *p*-value < 0.001) and lactating women (rho = 0.18; *p*-value < 0.001; Figure 3A). We did not observe any correlation between gestational age and AP-Q in pregnant women (Figure 3B), while there was positive and significant correlation between weeks of postpartum and AP-Q score (rho = 0.17; *p*-value < 0.001; Figure 3C).

### 3.3. AP-Q Categories

Next, we compared the degree of adherence to the HFP and the different categories defined in the AP-Q between different groups and the results are shown in Table 2.

Compared to the NPNL group, the pregnant and lactating groups showed significantly lower physical activity category. However, it was significantly higher in pregnant compared to lactating women. Within the healthy habits’ category, pregnant women scored significantly higher compared to the NPNL and the lactating group. The different dimensions within the healthy habits’ category were analyzed separately. Lifestyle was significantly lower in the NPNL group compared to pregnant and lactating women, while no difference was observed between pregnant and lactating women. Both, pregnant and lactating women showed significantly higher score for emotional balance compared NPNL; however, these groups did not show significant differences between them. In the sleep hygiene dimension, the NPNL group scored significantly higher followed by pregnant women and then the lactating women. All three groups were significantly different from each other. In culinary techniques, pregnant women scored significantly higher compared to the lactating and NPNL groups, but there were no statistical differences between pregnant and lactating groups. For the hydration category, pregnant and lactating women had similar and significantly higher than the NPNL group. Within the hydration category, other dimensions such as water intake, consumption of soft drinks, and wine and beers were not significantly different across the groups. As expected, in contrast to the NPNL group, pregnant and lactating women did not consume distilled beverages.

Concerning the dietary pattern assessed by APQ, in the grains, seed and legumes category, there was no significant difference across the groups. Regarding fruit consumption, pregnant women had significantly higher score compared to NPNL and lactating group while there were no statistical differences between the NPNL and lactating women. In vegetable intake, lactating women scored significantly higher than NPNL and pregnant women, but no statistical difference was seen between the NPNL and the pregnant groups. In the oil type category, no significant differences were detected between groups. In the dairy products category, pregnant women had a significantly higher score compared to the NPNL and the lactating groups, while the NPNL and the lactating groups did not show statistical differences. The animal protein intake category was not significantly different across the groups. In snack consumption, pregnant and lactating women had similar and significantly lower scores compared to the NPNL group (Table 3).

The regression models were used to evaluate the association between categories and AP-Q score in this cohort, considering NPNL as the reference group. A positive and significant association was found between the AP-Q score and pregnant women, but not in the lactating group. The higher AP-Q scores obtained by pregnant women indicate that they had better adherence to the HFP recommendations, being healthy habits, hydration, fruits, and snacks consumption the categories which most contributed to these differences. Besides, our data showed that physical activity was negatively associated with the lactating women, indicating that women significantly decreased their physical activity during lactation (Table 4).

## 4. Discussion

In this study, we aimed to analyze the degree of adherence to HFP, which represents a healthy dietary pattern, to assess various socio-economic factors that may affect adherence to healthy nutrition and lifestyle in Spanish women. We studied three groups of women, non-pregnant/non-lactating, pregnant, and lactating women, to detect which of these groups requires more attention to develop or implement nutritional or lifestyle programs. We found moderate adherence to the HFP in our cohort, and which is modulated by socioeconomic factors, with more adherence in older women and those of Spanish origin, women with higher educational levels and better economic status. The study also showed that breastfeeding women had the lowest adherence to the HFP, suggesting the need for specific interventions in this group.

In Spain, the age of childbearing has been gradually increasing over the last decades between 30 and 40 years old, with an average of 31.1 years [27]. Spain is one of the countries with the highest maternity age compared to other European countries [28]. This delay in the maternity age has already been observed in other studies and is associated with socioeconomic factors [29]. In support of these observations, we detected the median age of pregnant women to be 32 years and those of lactating women 34 years, while the NPNL group was younger (27.0 years). Most of our cohort was from university studies, were employed and making between 1000 to 2500 €/month for living. Educational level, financial income, and employment status are also in accordance with the Spanish social condition [30].

The mean AP-Q score was 6.09 out of 10 in our cohort, suggesting that women had a moderate adherence to the HFP. Several studies have shown a progressive decrease in adherence to the Mediterranean Diet in the general adult population, particularly in the new generations and in the intake of fruits, vegetables, and fish [31]. Pregnant women had the highest AP-Q score, compared with NPNL and lactating women, suggesting that pregnant women are more aware of their health status. It is also possible that they have greater support and follow-up by health professionals. In Spain, there are nutritional guidelines for pregnant women [3], but there are no specific guidelines for the breastfeeding period. Follow-up visits for pregnant women, where dietary and lifestyle guidelines are provided, promote self-care and healthy habits. This may also facilitate the adherence to the HFP recommendations, which include not only a healthier diet but also physical activity or emotional balance, among others. Other studies have shown that pregnant women have high adherence to the Mediterranean Diet. However, higher adherence was observed in women of higher socioeconomic status [32]. In contrast, with previous results showing lactating women with high adherence scores to the Mediterranean Diet [33], our results showed that lactating women obtained the lowest AP-Q score. However, in the Cuervo et al. study, women were recruited by attending pharmacies, and in our study, by social networks. Furthermore, the high AP-Q score could be due to different perspectives of lifestyle spheres such as social differences between pregnant and breastfeeding women. The overall perception in the population is that pregnant women need to take care of themselves and usually have more family support. After the delivery, care is focused on the newborn, with less focus on maternal health and self-care. Besides, in Spain, maternity leave is 16 weeks [34], which represents an additional difficulty in maintaining health habits and dietary patterns. These factors may explain the poorer adherence to Mediterranean Diet in breastfeeding mothers and suggest the need to improve follow-up and counseling for lactating women.

The AP-Q allows the assessment of several aspects related to adherence to the HFP under different dimensions. As expected, NPNL women did more exercise while breastfeeding women did least. This agreed with previous studies showing that physical activity decreases during pregnancy and continues to decrease during lactation [35]. NPNL women have no biosocial or physical impediment to physical exercise, while pregnant and lactating women have difficulties due to additional gain in body weight during pregnancy and puerperal recovery. In addition, breastfeeding women have less free time to practice exercise. In terms of healthy habits, including sleep hygiene and culinary techniques, pregnant women scored higher than the other groups, while breastfeeding women scored the least. As expected, breastfeeding women have the worse quality of sleep, followed by pregnant women, and the best were the NPNL women. In pregnant women, sleep quality may be reduced due to discomfort during the night particularly at later stages of gestation, and in lactating women since they have to breastfeed their children several times during day and night. It must be noted that sleep quality is not usually assessed in the context of HFP adherence, which is mostly focused on nutrition. However, it is a relevant aspect, particularly during breastfeeding, since the relationship of poor sleep quality with depression and emotional status has been demonstrated in other studies, especially in women during the puerperium period [36,37].

In our cohort, pregnant and lactating women had high adherence to the HFP recommendations in the hydration category, which could be partly related to the lack of distilled beverages consumption in these groups. There is insufficient evidence to support that an increased fluid intake is required by pregnant and lactating women to meet their physiological needs. In fact, there is controversy related to water intake requirements during lactation. Some data showed that water intake does not significantly affect the volume of breast milk produced [38]. Other studies concluded that lactating or pregnant women need to increase water intake to cope up with their physiological demands [39]. However, the effect of additional fluid intake, such as tea or coffee, is still unknown, due to insufficient systematic clinical/observational studies [40].

The grains, seed and legumes category showed moderate adherence to the HFP, with no significant differences across the three groups [41]. However, the consumption of legumes in Spain has decreased by more than 60% in recent years, even though all diets and nutritional recommendations include them [42]. The animal proteins category, which includes the consumption of eggs, meat, fish, and seafood, also showed moderate adherence with no significant differences across the groups. In contrast, the fruit category showed high adherence to the HFP, being higher in pregnant women reflecting a higher fruit consumption in Spain, which could be associated with adequate levels of water-soluble vitamins intake, as shown in our previous study in breastfeeding women [23]. In the vegetable category, even though lactating women had moderate adherence, they were higher than the pregnant and NPNL groups.

Consumption of extra virgin olive oil increases the score of the oil type category. In this category, our cohort had high adherence to the HFP in all three groups. The consumption of olive oil is widespread in the Spanish population and constitutes one of the most important sources of fat. The use of extra virgin olive oil in the Mediterranean Diet is associated with multiple positive outcomes on health, including immune and inflammatory responses [41]. The beneficial effect of extra virgin oil could be due to the presence of high bioactive compounds, including monounsaturated and omega-3 fatty acids [43], as well as vitamins A, E, and K [44].

Dairy products were the category that showed the lowest adherence to the HFP in all groups. Dairy products are considered an important source of calcium and vitamin D and are particularly important during pregnancy and lactation [45]. In fact, we have demonstrated previously that vitamin D intake in lactating women is below the recommendations and 24.0% of women had vitamin D deficiency in the first month of lactation [23]. Furthermore, in another European country, a high risk of vitamin D deficiency in breastfeeding women was found [46]. Sosa and Gomez de Tejada extensively reviewed the management of vitamin D deficiency [47]. These data suggest that improving vitamin D status in pregnant and lactating women should be included in the priorities of healthcare professionals and guidelines [48]. Nutritionists should consider vitamin D supplementation for pregnant women as soon as possible and, at least, supplement 37.5–50.0 µg/day from the second trimester of pregnancy. In addition, serum 25-hydroxyvitamin D levels should be monitored periodically to define the optimal dose and to verify the efficacy of supplementation [48].

The snacks category contains “junk food” that scores negatively because of higher levels of saturated fats, sugars, refined flours, and salt [49,50]. The high score in this category in all groups studied indicates the awareness of the population regarding the negative impact of snacks on health, which has prompted the food industry to develop healthier products.

This study demonstrates the lack of adherence to the HFP in breastfeeding mothers. Besides the fact that they have less time for their own health care, they also receive less counseling. Currently, nutritional interventions in this group are limited. It would be of great relevance to study their dietary pattern in more depth to develop specific guidelines for this vulnerable population that could be used by nutritionists and health professionals. In this regard, the AP-Q provides, a reliable, useful, and comprehensive tool that could be used to personalize nutritional intervention evaluating the particular scores gained in each category. Adequate counseling of breastfeeding mothers is relevant as it will have beneficial effects on both, the health of the mother and the newborn. Besides, nutritional education is a key tool to impact the health of future generations and increasing adherence to the Mediterranean diet.

In the Spanish public health system, it is mainly nurses and physicians who are responsible for nutritional education in society. However, the inclusion of dietitians and nutritionists in the public health system and multidisciplinary work between health professionals could lead to a significant improvement in these interventions. This study shows that nutritional interventions are currently not very effective and that it would be desirable to have nutrition professionals to increase the effectiveness of a given intervention. Adequate and personalized guidelines, accompanied by effective nutritional intervention, can improve postpartum recovery, maternal health, and influence breast milk composition [51,52], and thus can improve the short- and long-term health of the newborn [53,54].

A limitation of the study includes the two years of breastfeeding period for breastfeeding mothers. Considering the specific time of lactation period could help us understand, the most critical stage of breastfeeding, susceptible of nutritional interventions. The second limitation is the use of self-questionnaires. Even though they are useful, we are aware that, in general, when filling out nutritional questionnaires, the population tends to overestimate the intake of foods considered healthy and underestimate those considered less healthy.

In this study, we observed a strong influence of socioeconomic and demographic factors on adherence to HFP. It is possible that, women from the NPNL group, with a lower age range than the other groups, could have lower socioeconomic status. This could exert an influence on the adherence to HFP. This possibility remains open and could be analyzed in other studies, assessing the influence of age and economic factors on healthy habits, focused on the general population.

Furthermore, the AP-Q is helpful, not only to assess the cultural effect and detect women who are at risk from a nutritional point of view, susceptible to nutritional interventions and healthy habits, but also to carry out personalized interventions focused on the most required nutritional areas. The nutritional deviations observed in this study reaffirm the need to carry out specific health-promoting interventions using nutrition and healthy habits in the breastfeeding population, which would further improve the adjustment to the nutritional recommendations for this stage of the lifecycle.

## 5. Conclusions

Women in our cohort had a moderate adherence to the HFP, with younger women having less healthy lifestyles.Educational level, employment status, and high monthly income have a positive influence on adherence to the HFP.Breastfeeding women had the lowest adherence to nutritional, lifestyle, and healthy habits, suggesting the need for specific counseling.Dairy product consumption was low in the population, and deserves close attention, particularly in pregnant and breastfeeding women.Adherence to the recommendations on fruit, vegetable, and olive oil intake was adequate.It would be desirable to create guidelines of nutritional and lifestyle recommendations for breastfeeding women. Furthermore, follow-up by a nutritionist could improve women’s health during the puerperium.

## Figures and Tables

**Figure 1 nutrients-13-02372-f001:**
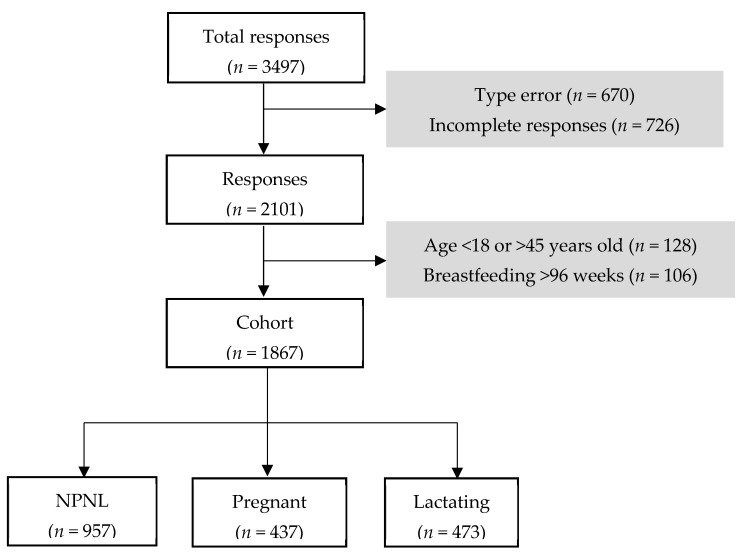
Flowchart diagram of enrollment. NPNL = non-pregnant/non-lactating women, *n* = sample size.

**Figure 2 nutrients-13-02372-f002:**
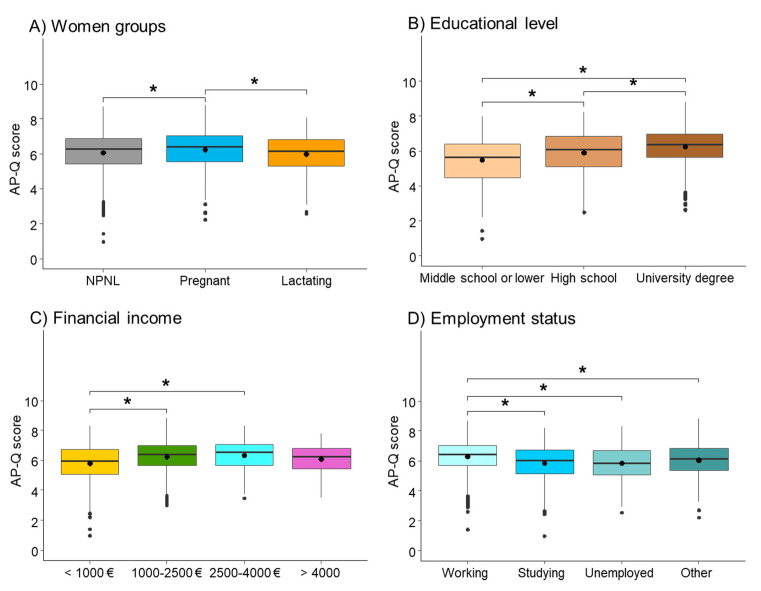
Comparison of Adherence to the Healthy Food Pyramid questionnaire (AP-Q) score between women groups (**A**), educational level (**B**), financial income (**C**), and employment status (**D**). Data has been shown as a box plot showing median and interquartile range. Mean is shown as a center dot. * *p*-Value < 0.05 by Dunn’s pairwise comparison to adjust for multiple comparisons. NPNL = non-pregnant/non-lactating.

**Figure 3 nutrients-13-02372-f003:**
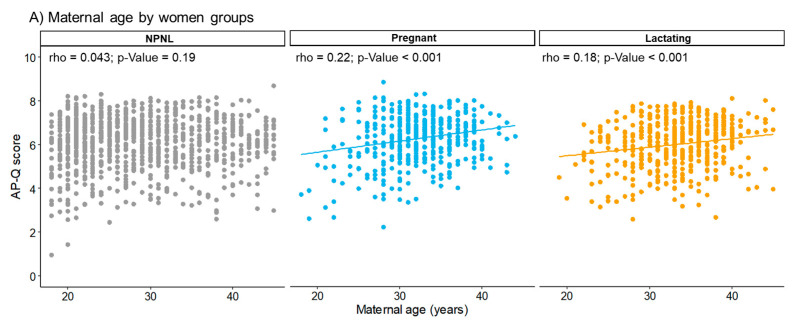
Correlation between Adherence to the Healthy Food Pyramid questionnaire (AP-Q) score and maternal age by women groups (**A**), and gestational age in pregnant women (**B**), and postpartum age in lactating women (**C**). *p*-Value was extracted by Rho-Spearman correlations. Linear trend is shown in significant correlations. NPNL = non-pregnant/non-lactating.

**Table 1 nutrients-13-02372-t001:** Cohort sociodemographic characteristics by groups.

	NPNL (*n* = 957)	Pregnant (*n* = 437)	Lactating (*n* = 473)	*p*-Value
Maternal age (years)	27.0 [22.0; 34.0] ^a^	32.0 [29.0; 35.0] ^b^	34.0 [30.0; 37.0] ^c^	<0.001
Origin				
Spanish	91.8% (879)	89.0% (389)	85.6% (405)	0.001
No Spanish	8.2% (78)	11.0% (48)	14.4% (68)	
Educational level				
Middle school or lower	5.8% (55)	7.8% (74)	11.6% (55)	<0.001
High school	32.8% (314)	16.9% (74)	18.4% (87)	
University degree	61.4% (588)	75.3% (329)	70.0% (331)	
Financial income				
≤1000€	47.9% (458)	24.9% (109)	29.2% (138)	<0.001
1001–2500€	46.5% (445)	64.5% (278)	58.6% (277)	
2501–4000€	4.8% (46)	9.2% (40)	11.0% (52)	
≥4001€	0.8% (8)	1.4% (6)	1.3% (6)	
Employment status				
Working	55.0% (526)	38.7% (169)	40.0% (189)	<0.001
Studying	32.4% (310)	4.4% (19)	2.8% (13)	
Unemployed	7.7% (74)	11.7% (51)	17.1% (81)	
Other situations	4.9% (47)	45.3% (198)	40.2% (190)	
Comorbidities	23.5% (225) ^a^	19.2% (84) ^b^	17.8% (84) ^b^	0.024

Data show median and interquartile range [Q1; Q3] in quantitative variables and relative frequency (%) and sample size (*n*) in qualitative variables. Values without a common letter significantly different, *p*-Value < 0.05. Quantitative variables were analyzed using the Kruskal-Wallis test, and qualitative variables using the Fisher’s exact test; NPNL = non-pregnant/non-lactating.

**Table 2 nutrients-13-02372-t002:** Comparison between groups in the physical activity, healthy habits, and hydration categories.

	NPNL (*n* = 957)	Pregnant (*n* = 437)	Lactating (*n* = 473)	*p*-Value
Physical activity	0.58 [0.31; 0.87] ^a^	0.48 [0.24; 0.76] ^b^	0.40 [0.19; 0.67] ^c^	<0.001
Healthy habits	0.64 [0.49;0.75] ^a^	0.68 [0.54;0.76] ^b^	0.61 [0.48;0.70] ^c^	<0.001
Lifestyle	0.80 [0.00; 1.00] ^a^	1.00 [0.60; 1.00] ^b^	1.00 [0.60; 1.00] ^b^	<0.001
Emotional balance	0.67 [0.50; 0.80] ^a^	0.73 [0.60; 0.83] ^b^	0.70 [0.50; 0.83] ^b^	<0.001
Sleep hygiene	0.69 [0.46; 0.77] ^a^	0.54 [0.38; 0.69] ^b^	0.38 [0.23; 0.54] ^c^	<0.001
Culinary techniques	0.64 [0.50; 0.79] ^a^	0.71 [0.50; 0.86] ^b^	0.64 [0.50; 0.79] ^a^	0.021
Hydration	0.73 [0.56; 0.91] ^a^	0.80 [0.66; 0.96] ^b^	0.76 [0.64; 0.96] ^b^	<0.001
Water intake	1.00 [0.80; 1.00]	1.00 [0.80; 1.00]	1.00 [0.80; 1.00]	0.28
Soft drinks	−0.11 [−0.78; −0.11]	−0.11 [−0.62; −0.11]	−0.11 [−0.82; −0.11]	0.30
Wine & beers	0.0 [0.0; 0.0]	0.0 [0.0; 0.0]	0.0 [0.0; 0.0]	0.07
Distilled beverages	−0.17 [−0.17; 0.0] ^a^	0.0 [0.0; 0.0] ^b^	0.0 [0.0; 0.0] ^b^	<0.001

Data show median and interquartile range [Q1; Q3]; *p*-value was reported by Kruskal-Wallis test. Data labeled with a different letter represents statistically significant differences with a *p*-value < 0.05 by Dunn’s pairwise comparison to adjust for multiple comparisons. NPNL = non-pregnant/non-lactating.

**Table 3 nutrients-13-02372-t003:** Comparison between groups in the grains, seed and legumes, fruits, vegetables, oil type, dairy products, animal proteins, and snacks categories.

	NPNL (*n* = 957)	Pregnant (*n* = 437)	Lactating (*n* = 473)	*p*-Value
Grains, seed, and legumes	0.58 [0.36; 0.78]	0.60 [0.36; 0.79]	0.59 [0.35; 0.79]	0.62
Fruits	0.80 [0.40; 1.00] ^a^	1.00 [0.80; 1.00] ^b^	0.80 [0.40; 1.00] ^a^	<0.001
Vegetables	0.56 [0.47; 0.64] ^a^	0.56 [0.47; 0.64] ^a^	0.58 [0.50; 0.64] ^b^	0.018
Oil type	0.92 [0.67; 1.00]	1.00 [0.67; 1.00]	0.83 [0.58; 1.00]	0.20
Dairy products	0.37 [0.33; 0.43] ^a^	0.39 [0.33; 0.43] ^b^	0.37 [0.33; 0.41] ^a^	0.005
Animal proteins	0.54 [0.42; 0.62]	0.54 [0.46; 0.62]	0.54 [0.42; 0.62]	0.06
Snacks	0.72 [0.62; 0.80] ^a^	0.67 [0.62; 0.77] ^b^	0.70 [0.62; 0.77] ^b^	0.002

Data show median and interquartile range [Q1; Q3]; *p*-value was reported by Kruskal-Wallis test. Data labeled with a different letter represents statistically significant differences with a *p*-value < 0.05 by Dunn’s pairwise comparison to adjust for multiple comparisons. NPNL = non-pregnant/non-lactating.

**Table 4 nutrients-13-02372-t004:** Regression models between groups, categories, and the AP-Q score.

	Pregnant Women		Lactating Women	
	β ± SE	*p*-Value	β ± SE	*p*-Value
AP-Q score	0.26 ± 0.07	0.001	−0.02 ± 0.06	0.80
Physical activity	−0.03 ± 0.02	0.13	−0.11 ± 0.02	<0.001
Healthy habits	0.07 ± 0.01	<0.001	0.0 ± 0.0	0.85
Hydration	0.07 ± 0.02	<0.001	0.04 ± 0.01	0.005
Fruits	0.11 ± 0.02	<0.001	0.04 ± 0.02	0.027
Vegetables	0.0 ± 0.01	0.89	0.02 ± 0.01	0.015
Dairy products	0.0 ± 0.0	0.18	0.0 ± 0.0	0.19
Snacks	−0.02 ± 0.01	0.028	−0.01 ± 0.01	0.027

Data show estimated beta (β) coefficients ± standard error (SE) and *p*-Value. Non-pregnant/non-lactating group was considered as the reference group. Models were adjusted by maternal age, origin, educational level, financial income, employment status, and comorbidities.

## Data Availability

The data presented in this study are available on request from the corresponding author. The availability of the data is restricted to investigators based in academic institutions.

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
