# Peer review of "Assessment of Adherence to the Healthy Food Pyramid in Pregnant and Lactating Women"

_nutrients, 2021, doi:10.3390/nu13072372_

Round 1

Reviewer 1 Report

Andrea Gila-Díaz et colleague  investigated the differences in adherence to the HFP in pregnant, lactating, and non-pregnant/non-lactating women concerning sociodemographic factors using AP-Q to assess adherence to the HFP. Authors have alrealdy published other work in this field on this journal so they have adequately described and clearly presented their results and conclusion.

Author Response

Andrea Gila-Díaz et colleague investigated the differences in adherence to the HFP in pregnant, lactating, and non-pregnant/non-lactating women concerning sociodemographic factors using AP-Q to assess adherence to the HFP.

Authors have already published other work in this field on this journal so they have adequately described and clearly presented their results and conclusion.

Response: Thank you so much for your words. We appreciate your time dedicated reviewing our work.

Reviewer 2 Report

The study is important because it determines the group of breastfeeding women to which educational and other activities aimed to improving the diet and lifestyle should be devoted. However, it is incomprehensible to me, why the age range in NPNL group starts at 22 and in pregnant and lactating groups from 29-30, which entails socioeconomic differences - in education and income. I propose to recalculate the data by adjusting the age range of women NPNL to the age of pregnant and lactating women. Besides, I cannot agree with not including wine and beer as alcohol.

Author Response

The study is important because it determines the group of breastfeeding women to which educational and other activities aimed to improving the diet and lifestyle should be devoted.

Response: Thank you for your time reviewing our work. Please, kindly see our comments below.

However, it is incomprehensible to me, why the age range in NPNL group starts at 22 and in pregnant and lactating groups from 29-30, which entails socioeconomic differences - in education and income. I propose to recalculate the data by adjusting the age range of women NPNL to the age of pregnant and lactating women.

Response: The same inclusion criteria regarding age was the same for all the groups (between 18 and 45 years old; line 105). The fact that we had older women in the pregnant and lactating groups, probably reflects the social change in Spain, which has gradually increased childbearing age. Therefore, we think that including all the data from NPNL women reflects better the social reality of the Spanish population. We understand your point and agree on the fact lower age of NPNL women may have influenced the socioeconomic status. Although it is possible to hypothesize that older NPNL women could have a better adherence to HFP due to higher socioeconomic status, the statistical analysis showed in NPNL group, AP-Q score was not correlated with age, while it was positively correlated in pregnant and lactating women. Therefore, we think that eliminating the younger age group of NPNL women would not modify the results and conclusions of the study. Besides, maternal age, educational level, financial income and employment status were parameters used to adjust the models.

However, the review’s comment is very appropriate and we have included this aspect in the discussion (lines 448-452), since we think it deserves further studies focused on the impact of age and socioeconomic factors on healthy habits.

Besides, I cannot agree with not including wine and beer as alcohol.

Response: We agree that they contain alcohol. However, in lower quantity compared to alcohol content of distilled beverages (rum, gin, whiskey or similar). Besides, due to their chemical processing, fermented beverages provide macro- and micronutrients (such as vitamins, antioxidants or minerals) that distilled beverages do not (PMID: 16562818; PMID: 14679365). Thus, the HFP considers that fermented beverages can be moderately consumed, while distilled beverages are excluded from it, being their consumption preferably null (http://dietamediterranea.com/piramidedm/piramide_INGLES.pdf). Therefore, beers & wines and distilled beverages were separated in two different dimensions since the AP-Q measures nutritional behavior and lifestyles as a function of distance from the HFP. While all other dimensions and categories scored from 0 to 1, these two dimensions can score negatively (line 157). Both dimensions belong to the hydration category and could decrease its score, but in different ways. However, to avoid possible confusion, we have modified the text by replacing "alcohol" with "distilled beverages".